# Depleting *Mycobacterium tuberculosis* of the transcription termination factor Rho causes pervasive transcription and rapid death

Laure Botella[1], Julien Vaubourgeix[1], Jonathan Livny[2] & Dirk Schnappinger[1]

Rifampicin, which inhibits bacterial RNA polymerase, provides one of the most effective treatments for tuberculosis. Inhibition of the transcription termination factor Rho is used to treat some bacterial infections, but its importance varies across bacteria. Here we show that Rho of *Mycobacterium tuberculosis* functions to both define the 3′ ends of mRNAs and silence substantial fragments of the genome. Brief inactivation of Rho affects over 500 transcripts enriched for genes of foreign DNA elements and bacterial virulence factors. Prolonged inactivation of Rho causes extensive pervasive transcription, a genome-wide increase in antisense transcripts, and a rapid loss of viability of replicating and non-replicating *M. tuberculosis in vitro* and during acute and chronic infection in mice. Collectively, these data suggest that inhibition of Rho may provide an alternative strategy to treat tuberculosis with an efficacy similar to inhibition of RNA polymerase.

[1] Department of Microbiology and Immunology, Weill Cornell Medicine, 413E 69th Street, New York, New York 10021, USA. [2] Broad Institute of MIT and Harvard, 415 Main Street, Cambridge, Massachusetts 02142, USA. Correspondence and requests for materials should be addressed to D.S. (email: dis2003@med.cornell.edu).

Tuberculosis (TB), a communicable disease caused by *Mycobacterium tuberculosis*, killed 1.5 million people in 2014 (ref. 1). *M. tuberculosis* thus currently ranks ahead of HIV as the most frequent microbial killer of humans. Deaths due to TB are unlikely to decline soon as a third of the world's population is estimated to be infected and therefore at risk for developing TB. When infected with drug-susceptible (DS) *M. tuberculosis*, more than 95% of TB patients can be cured by treatment for two months with four drugs (rifampicin (RIF), isoniazid (INH), pyrazinamid and ethambutol) and with two drugs (RIF and INH) for the following four months[2]. However, its complexity and length make TB chemotherapy difficult to implement in resource-limited settings leading to high failure rates and the emergence of drug-resistant (DR) *M. tuberculosis*[3].

The two most important drugs for treating TB are INH, which has the most potent bactericidal activity during the early phase of treatment, and RIF, which most effectively prevents relapse[2,4]. TB caused by *M. tuberculosis* that is resistant to INH and RIF is classified as multidrug-resistant, irrespectively of resistance to other drugs, because of the importance of INH and RIF for TB chemotherapy. New TB drugs, which are urgently needed, would ideally share the properties that make INH and RIF the cornerstones of the current first-line TB chemotherapy. One strategy to identify such drugs is to search for new molecules that inhibit the same pathways as INH and RIF.

INH inhibits the synthesis of mycolates, which are essential for maintaining the integrity of the mycobacterial cell envelope[5]. Several new small molecules that interfere with synthesis of mycolates or the synthesis of other essential cell envelope components are being evaluated for their potential to replace or synergize with INH (refs 6,7). RIF inhibits bacterial RNA polymerases (RNAP). RNAP consist of a core, containing β, β′, ω, two α-subunits and a σ-factor that associates with the core enzyme for promoter recognition and transcription initiation[8,9]. Most inhibitors of RNAP interact with its enzymatic core and interfere with initiation and/or elongation of transcription[10]. Specifically, RIF binds to the β subunit and inhibits transcription initiation[11].

Bacterial viability depends not only on the accurate initiation and elongation of RNA synthesis but also on its appropriate termination. There are two mechanisms of transcription termination by bacterial RNAP. The RNAP core alone is sufficient to end RNA synthesis at so-called intrinsic terminators. Factor-dependent terminators require additional proteins to stop transcription[12]. The RNA/DNA helicase/translocase Rho is the main termination factor and is broadly conserved among bacteria. In *Escherichia coli*, Rho terminates transcription at ~1,300 loci, contributes to gene polarity, helps resolve R-loops to maintain genome stability, assures foreign DNA silencing, prevents pervasive antisense transcription and is required for normal growth in rich media[13–15]. Inhibition of Rho by the natural product bicyclomycin (BCM), which prevents ATP turnover by Rho, is toxic for many gram-negative bacteria and the gram-positive *Micrococcus luteus*[16–19]. In contrast, Rho is dispensable for normal growth in rich media for several other gram-positive bacteria including *Bacillus subtilis*[20,21], *Streptomyces lividans*[22] and *Staphylococcus aureus*[23]. Interestingly, Rho of *M. tuberculosis* is predicted to be essential for growth[24,25], but is poorly inhibited by BCM (ref. 26), which suggests that targeting mycobacterial Rho could allow development of genus-specific antibiotics.

Here, we apply a genetic approach to determine the consequences of inactivating Rho in *M. tuberculosis*. Using RNA-Seq, we identify ~300 loci and ~80 genes whose transcription is affected by Rho and find that depletion of Rho induces pervasive transcription across the entire genome of *M. tuberculosis*. We further demonstrate that *M. tuberculosis* requires enzymatically active Rho to grow in rich media, survive nutrient starvation *in vitro* and establish acute infection and maintain chronic infection in mice.

## Results

**Essentiality of Rho in *M. tuberculosis*.** We first constructed a merodiploid strain that carried a second copy of *rho* integrated into the attachment site of the phage L5 (attL5). This second *rho* was under transcriptional control of the 600 base pair region putatively encoding its native promoter ($P_{rho}$) (Supplementary Fig. 1). We then inactivated the wild type (WT) copy of *rho* by homologous recombination. This yielded *M. tuberculosis* Δ*rho*::$P_{rho}$*rho*, which contains a single copy of *rho*, integrated into attL5. Plasmids integrated into attL5 can be exchanged with high frequency by site-specific recombination[27]. Δ*rho*::$P_{rho}$*rho* thus allowed us to delete *rho* by transformation with another plasmid that integrated into attL5 in place of the *rho*-encoding plasmid and did not contain Rho (empty vector). These empty vector transformations did not result in colonies on standard agar plates (Fig. 1a). In contrast, many colonies grew after transformation with a plasmid that reintroduced a functional copy of *rho* into the genome (transcribed either by its native promoter or the strong hsp60 promoter; Fig. 1a). We concluded that *M. tuberculosis* requires Rho to grow on standard agar plates.

The essentiality of Rho prompted us to construct strains in which Rho expression can be silenced with anhydrotetracycline (atc)[28]. First, we replaced *rho* of Δ*rho*::$P_{rho}$*rho* with a copy of *rho* transcribed by a strong promoter that can be repressed by a reverse Tet repressor (RevTetR)[29–31]. In this strain (Δ*rho*:PTetOFF-rhoSD2), *rho* was transcriptionally repressed on activation of RevTetR by atc. Intermediate concentrations of atc allowed growth of Δ*rho*:PTetOFF-rhoSD2 on solid medium, but neither high- nor low-concentrations of atc permitted colony formation (Fig. 1a). This suggested that both overexpression and depletion of Rho prevented growth of *M. tuberculosis*. In the second strain, we replaced *rho* of Δ*rho*::$P_{rho}$*rho* with a copy of *rho* (rhoDAS), which encoded the DAS + 4-tag at its 3′-end. This tag is recognized by SspB, an *E. coli* protein that can deliver DAS + 4-tagged proteins for degradation by ClpXP in mycobacteria[32,33]. Addition of the DAS + 4-tag reduced the *in vivo* activity of Rho in *M. tuberculosis* (Supplementary Fig. 2), but fully complemented Δ*rho* if expressed by a strong promoter (Fig. 1a). We then used Δ*rho*::$P_{TetOFF}$rhoDAS to generate a DUal Control *rho* strain (Rho-DUC), in which atc causes both transcriptional repression and proteolytic degradation of DAS + 4-tagged Rho (Supplementary Fig. 2)[34]. Rho-DUC quickly ceased to grow, when atc was added to the liquid medium (Fig. 1b). Western blots showed that Rho levels were reduced by ~50% 6 h after the addition of atc and Rho could not be detected 24 h after the addition of atc (Fig. 1c, Supplementary Fig. 12). The rapid depletion of Rho in the Rho-DUC mutant and the concurrent effect on growth conducted us to determine the importance of transcription termination in *M. tuberculosis*.

**Impact of depleting Rho on the transcriptome.** We defined the changes that occur in the transcriptome of *M. tuberculosis* after inactivating Rho by sequencing RNA isolated from Rho-DUC before addition of atc, and after cultivation with atc for 1.5, 3, 6 and 9 h. As controls, we sequenced RNA isolated from WT *M. tuberculosis* before addition of atc, and after growth with atc for 6 h. Very few transcripts significantly changed in abundance after exposing Rho-DUC to atc for 1.5 or 3 h (change >2.5 and False Discovery Rate <0.05, Supplementary Data 1), but exposure to atc for 6 h changed 507 transcripts (Supplementary

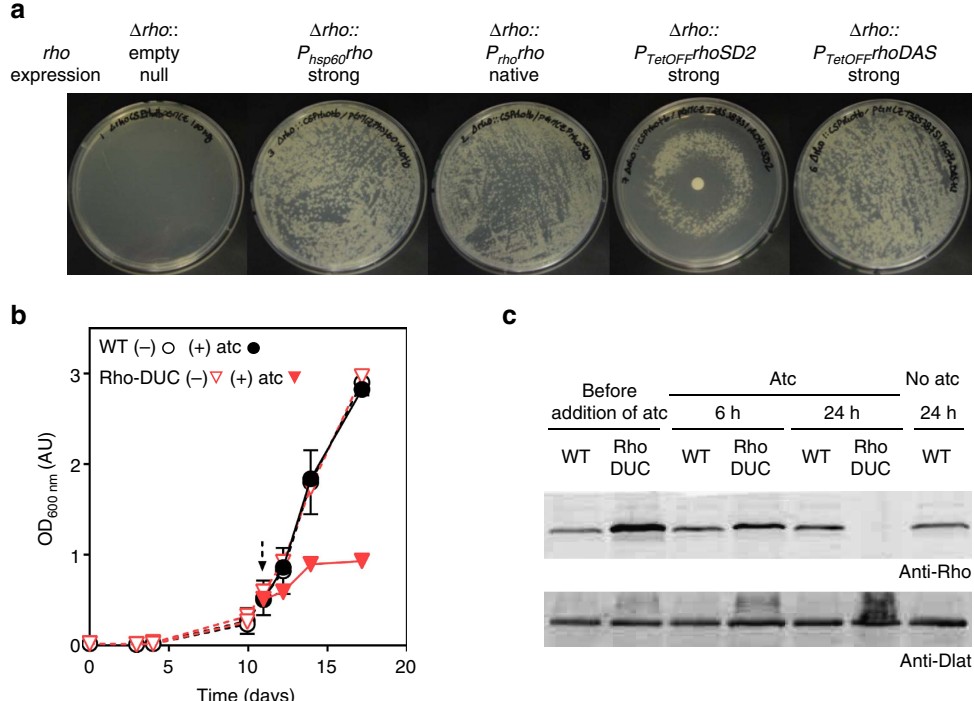

**Figure 1 | The gene *rho* is essential for the growth of *M. tuberculosis*.** (**a**) Phenotypes of different *rho* mutants on rich solid medium. Rho was either deleted (Δ*rho*) or exchanged against *rho* expressed under the control of the *hsp60* promoter ($P_{hsp60}rho$), the native promoter ($P_{rho}rho$) or transcriptionally regulated TetOFF promoters ($P_{TetOFF}rho$SD2, $P_{TetOFF}rho$DAS). When applicable, 100 ng of atc was added on a disc placed at the centre of the agar plate. (**b**) Growth of WT and Rho-DUC Mtb strains in the presence or absence of atc. The arrow indicates when 1 µg ml$^{-1}$ atc was added. Data are means of duplicate (WT) or triplicates (Rho-DUC) cultures representative of five experiments (± s.d., some error bars too small to be seen). (**c**) Western blot to detect the Rho protein in the lysates from cultures shown in (**b**). The entire blots are shown in Supplementary Fig. 12.

Fig. 3a, Supplementary Data 1). In contrast, the abundance of only 26 transcripts was significantly altered in WT *M. tuberculosis* after 6 h of atc treatment (Supplementary Fig. 3b). This indicated that the vast majority of changes observed in Rho-DUC were due to depletion of Rho and not caused by atc itself. Variations in antisense transcripts were more abundant than changes in sense transcripts and there was no correlation between the levels of sense and antisense transcripts (Supplementary Fig. 3c). Thus, the increase in transcription of antisense RNAs did not affect the levels of sense transcript.

In analogy to the bicyclomycin-significant transcripts defined by chemical inhibition of Rho in *E. coli*[14], we refer to regions of the *M. tuberculosis* chromosome whose representation in the transcriptome changes directly in response to genetic inactivation of Rho as Rho-significant regions (RSRs). We reasoned that reduced Rho-dependent transcription termination should be a primary effect of Rho depletion and occur before the transcriptome changed more broadly. We anticipated that a failure of transcription termination would increase transcript length (Supplementary Fig. 4a). Therefore, we focused our analyses on the 6 h time point when 96% of the changes affecting sense transcripts stemmed from genomic regions that were underrepresented in the transcriptome before Rho depletion (Supplementary Fig. 3a). By definition, RSRs cannot identify Rho-dependent terminators directly; but they nevertheless give an indication on the location of Rho-dependent terminators by documenting, which part of the transcriptome change in response to inactivation of Rho.

We first analysed regions of the *M. tuberculosis* chromosome that showed similarly high-transcript levels in samples that were isolated after Rho had been depleted for 6 h and the 6 h controls. These regions of equal transcript representation were chosen to

enrich for areas that were actively transcribed irrespectively of the expression of Rho and to select against new transcripts initiated in response to Rho depletion. We then scanned the transcriptome to identify those regions of equal transcript representation that were followed by a segment in which transcript abundance declined in the reference samples yet remained high (or increased) in the Rho-depleted sample. We evaluated several numerical filters and reproducibility criteria for the RSR identification strategy (as described in Supplementary Figs 4b and 5a) and chose criteria that defined a total of 303 RSRs (Supplementary Data 2). These RSRs are evenly distributed across the entire *M. tuberculosis* genome and located on the minus and the plus strands at similar frequencies (155 and 148, respectively) (Fig. 2a). Examples of these RSRs are shown in Fig. 2b,c. The majority (78%) of RSRs are located in intergenic regions (Supplementary Data 2) and many of them occur close to the 3′-end of genes located on the same strand as the RSR (Supplementary Fig. 5c). This indicates a prominent role for Rho in the transcription termination of protein encoding genes. No bias was introduced by transcript levels as the distribution of RSRs was independent of transcript abundance (Supplementary Fig. 5b).

In contrast to Rho-dependent transcription termination, intrinsic termination is often abrupt and strong Rho-independent terminators generally have relative precise stop points[35]. We therefore also analysed our RNA-Seq data for regions in which transcription terminated rapidly (as described in Methods). This analysis identified 191 regions, equally distributed between the minus and plus strands (96 and 95, respectively) (Supplementary Data 4). In addition, 44 of these had previously been predicted to contain Rho-independent terminators by different computational methods including those used by Mitra and colleagues[36] and/or

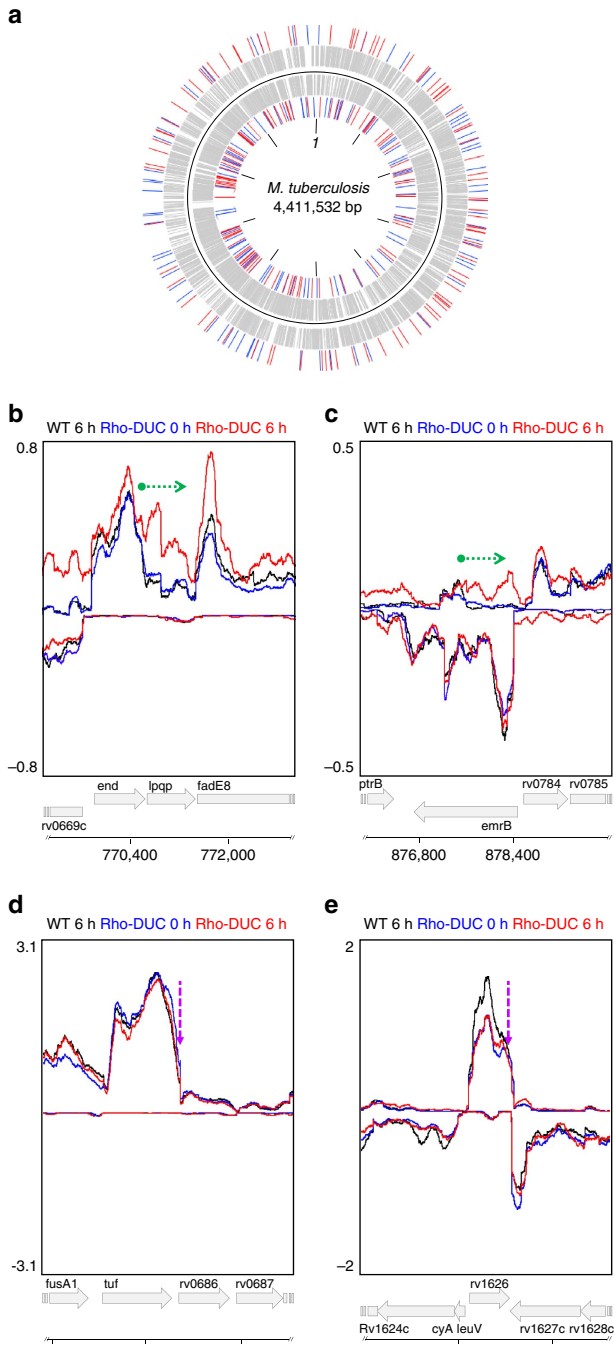

**Figure 2 | RSRs affect sense and antisense transcription across the genome.** (**a**) Distribution of the RSRs in the genome of *M. tuberculosis*. ( + ) and ( − ) strand features are depicted respectively on the outside and on the inside of the black solid line. Grey bars represent the genes, while the RSRs are indicated by blue (silencers) and red (transcripts terminators) bars. (**b,c**) Examples of RSRs visualized in the Artemis genome browser. (**d,e**) Examples of putative Rho-independent terminators visualized in the Artemis genome browser. The curves represent the average number of reads of replicates of the WT strain at 6 h of atc (black), and the Rho-DUC strain right before addition (0 h, blue) or 6 h after the addition of atc (red). The top half shows the traces obtained for the ( + ) strand and the bottom half the traces of the ( − ) strand. The green arrow indicates the RSR's direction and length. The purple arrows indicate a putative Rho-independent terminator identified.

Gardner and colleagues[37]. Depletion of Rho did not prevent termination in these regions, which strongly suggest that they reflect Rho-independent termination of transcription (see Fig. 2d,e for three examples). Futhermore, this analysis confirmed that depletion of Rho did not cause a general defect in RNAP function and allowed to accurately distinguish between Rho-dependent and Rho-independent transcription termination.

Some bacteria utilize Rho not only to define the 3′-ends of messenger RNAs (mRNAs) but also to silence regions of the genome[13,14]. To determine if Rho also functions as a genome silencer in *M. tuberculosis*, we classified RSRs as silencing elements if they were not located in the vicinity of an open reading frame (ORF) encoded on the same strand or covered > 50% of an ORF encoded on the same strand as the RSR. These criteria led us to classify 44.5% of RSRs as silencers (Supplementary Data 2 and 3). They are distributed along the genome (Fig. 2) and many of them overlap, at least partially, with genes encoded on the opposite strand. ORFs that were silenced by Rho included many genes related to insertion elements and phages, suggesting that Rho silences foreign DNA in *M. tuberculosis* (Fig. 3, Supplementary Data 3,5 and 6). In addition, 10% of the genes silenced by Rho during growth in a rich medium encode PE/PPE proteins, when this group of genes only represents 4% of all *M. tuberculosis* genes. PE/PPE proteins are unique to mycobacteria, are particularly abundant in pathogenic mycobacteria, and are defined by Pro-Glu (PE) and Pro-Pro-Glu (PPE) motifs near the N-terminus. Although their function is recondite, various PE/PPE proteins are required for mycobacterial virulence[38]. In contrast, examining the annotation of the 171 genes, whose transcription was terminated by an adjacent RSR (Supplementary Data 7) did not reveal enrichment towards a particular functional category (Fig. 3).

To determine the impact of prolonged Rho-depletion we analysed RNA isolated from Rho-DUC after exposure to atc for 9 and 24 h, respectively. Rho-depletion of 9 h led to five times more transcripts variations than 6 h (2,577 transcripts at 9 h depletion for 507 changes observed after 6 h), and caused a relative increase of antisense over sense transcription; these changes were exacerbated after 24 h of Rho depletion (Supplementary Figs 3a and 6, Supplementary Data 1). Prolonged depletion of Rho thus caused transcriptional accuracy to collapse and RNA synthesis to continue, when it would have otherwise been halted, thus leading to the accumulation of pervasive antisense transcripts. We predicted this to severely impact *M. tuberculosis* viability, likely because of an increased R-loops formation[15], and the failure to maintain chromosome integrity[39].

**Cidality of *rho* silencing *in vitro* in *M. tuberculosis*.** Depletion of Rho in standard liquid media did not impair viability of *M. tuberculosis* if it occurred for only 6 h, but colony forming units (CFU) decreased drastically after longer periods of depletion (Fig. 4a; Supplementary Fig. 7). Exposure of Rho-DUC to atc for 24 h decreased CFU by approximately five orders of magnitude (Fig. 4a) and had dramatic effect on growing *M. tuberculosis*. We next aimed to determine the consequences of Rho inactivation on non-replicating *M. tuberculosis* and therefore characterized Rho-DUC during nutrient starvation. In this condition, CFUs of Rho-DUC also decreased rapidly and drastically after the addition of atc (Fig. 4b). To rule out that depletion of *rho* only prevented growth in the presence of atc we constructed Rho-TetON (Δ*rho::P_{tb38}-rhoDAS + 4::sspB-TetON*), in which degradation of Rho is prevented by atc and induced by cultivation in the atc-free media. *M. tuberculosis* Rho-TetON was only able to grow in the presence of atc and ceased to grow when atc was removed (Supplementary

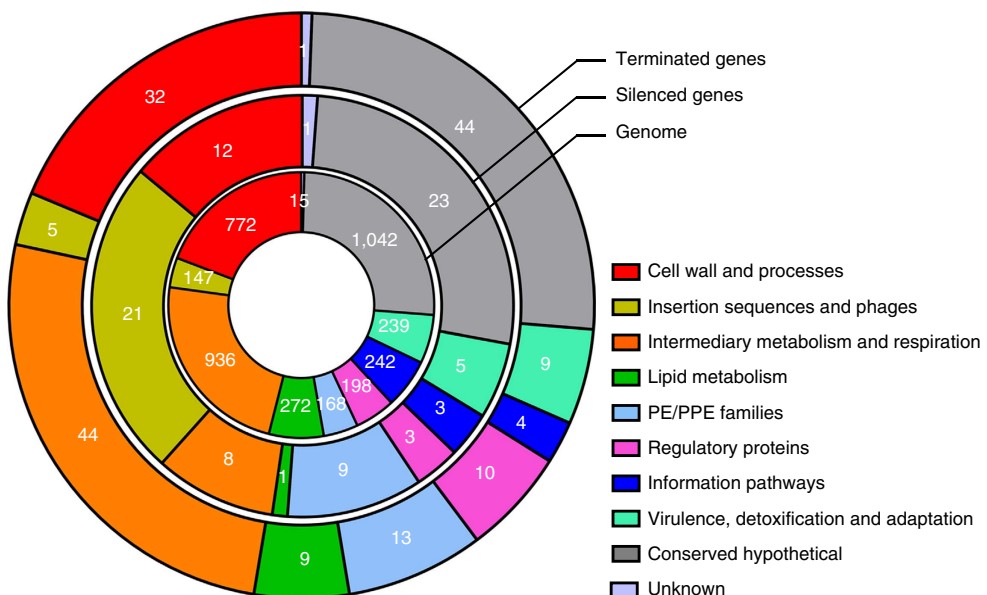

**Figure 3 | Genes silenced or terminated by Rho in *M. tuberculosis*.** Classification of *M. tuberculosis* genes according to their function and impact of Rho. The genes silenced or terminated by Rho are classified in families based on their product function (Tuberculist). The percentage of genes in each category is represented as sections of the circle. The repartition of all genes of the *M. tuberculosis* genome in the same functional categories has been added for comparison. The number in white represents the absolute number of genes found for each category.

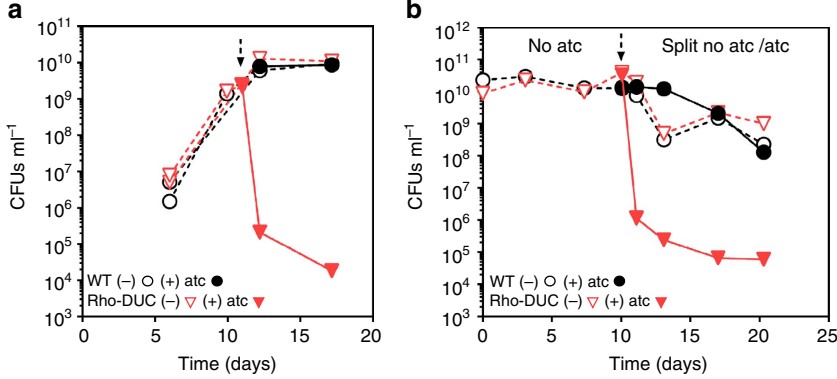

**Figure 4 | Depleting Rho kills replicating and non-replicating *M. tuberculosis*.** Impact of Rho deletion by addition of atc on the viability of WT and Rho-DUC *M. tuberculosis* strains during growth in liquid media (**a**) and starvation in PBS (**b**). The dotted arrows indicate when atc was added. In (**a**) data are means of representative duplicate (WT) or triplicates (Rho-DUC) cultures of an experiment that was performed five times ( ± s.d., some error bars are too small to be seen). The experiment in (**b**) is representative of four experiments.

Fig. 8). Collectively, these results demonstrated that Rho is required for growth and survival of *M. tuberculosis* in rich media, and for survival during starvation-induced non-replicating persistence.

**Importance of the Rho's ATP binding motif**. All Rho homologues contain conserved RNA binding domains and multiple ATP hydrolysis signature motifs. In addition, approximately a third of Rho proteins, including Rho from *M. tuberculosis*, contain a large extra N-terminal sequence of unknown function[40]. The first experiments that analysed transcription termination by *M. tuberculosis* Rho suggested it to not require ATP hydrolysis to terminate transcription[41]. It was hypothesized that the N-terminal domain might be responsible for the apparent dispensability of ATP hydrolysis. In contrast, a second study found ATP hydrolysis to be essential for transcription termination by *M. tuberculosis* Rho *in vitro*[26]. To evaluate importance of ATP binding to Rho for growth of *M. tuberculosis* we mutated the amino acids D440, R541 and E386 to N440, A541

and A386. D440 and R541 are located in the Walker B motif and the arginine finger, respectively, and mutation of the corresponding amino acids in Rho of *E. coli* (D265 to N265 and R366 to A366) dramatically reduced ATP hydrolysis without affecting the binding of Rho to RNA (ref. 42). In addition, a point mutant of the catalytic glutamate E386 of *M. tuberculosis* Rho failed to hydrolyse ATP, to unwind duplex and to terminate transcription *in vitro*[26]. Unlike WT Rho, Rho-D440N, Rho-R541A or Rho-E386A did not sustain growth of Rho-DUC with atc (Fig. 5a,b; Supplementary Fig. 9a,b). To verify that the absence of complementation was not due to the lack of expression of mutated Rho, we also constructed FLAG-tagged versions of WT Rho and the three-point mutants. Similar protein levels were detected for all three constructs (Fig. 5c, Supplementary Fig. 13, Supplementary Fig. 9c, and Supplementary Fig. 14). This demonstrated that the *in vivo* function of *M. tuberculosis* Rho depends on D440, R541 and E386 and strongly suggests that Rho-dependent transcription termination in *M. tuberculosis* requires ATP hydrolysis.

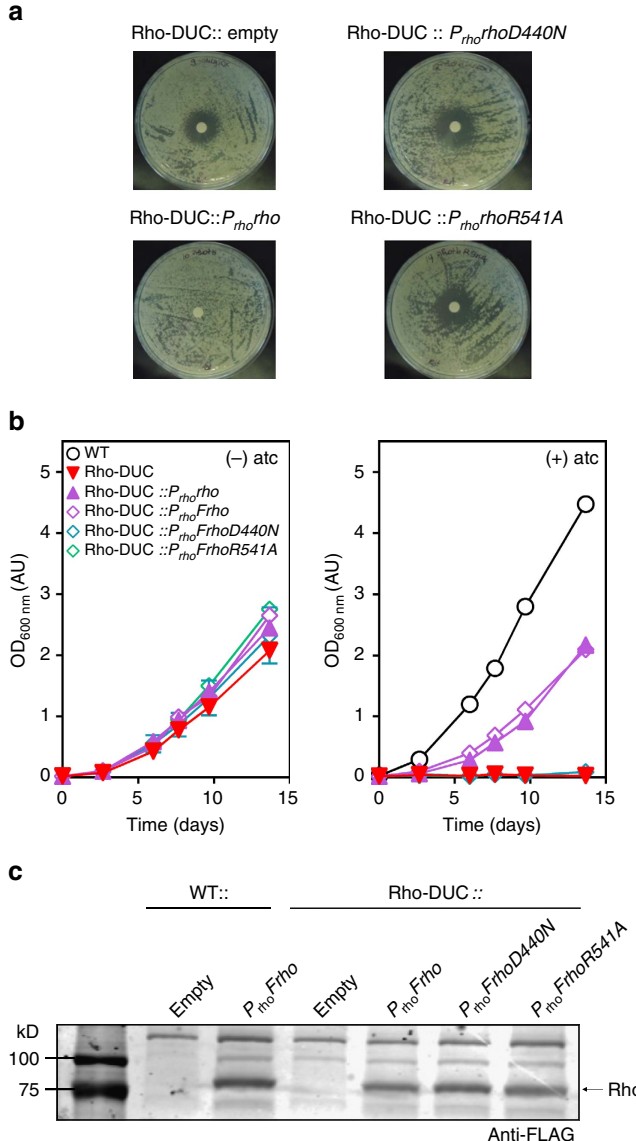

**Figure 5 | Importance of ATP binding and hydrolysis for Rho.**
(**a**) Complementation of Rho-DUC on solid medium. The growth inhibitory effect of Rho depletion by addition of atc was assayed on the Rho-DUC strain that constitutively expresses a second copy of WT Rho ($P_{rho}rho$), or Rho-D440N or Rho-R541A ($P_{rho}rhoD440N$ and $P_{rho}rhoR541A$). Approximately 100 ng of atc was added on the discs. (**b**) Complementation of Rho-DUC in liquid medium. The Rho-DUC strain constitutively expresses the second copy of WT Rho ($P_{rho}rho$), or a FLAG-tagged WT Rho ($P_{rho}Frho$), or a FLAG-tagged Rho-D440N or a FLAG-tagged Rho-R541A ($P_{rho}FrhoD440N$ and $P_{rho}FrhoR541A$). Data are means of duplicate cultures ( ± s.d., some error bars too small to be seen). (**c**) Western blot of protein lysates using an anti-FLAG antibody of samples grown in the absence of atc in (**b**). The entire blot is shown in Supplementary Fig. 13.

**Importance of Rho for *M. tuberculosis* in mice.** To determine the impact of silencing Rho on *M. tuberculosis in vivo* we infected mice by aerosol. Silencing of *rho* expression in Rho-DUC was achieved by supplementing mice food with doxycycline as described previously[34]. In absence of doxycycline, the number of CFU isolated from the lungs and spleens of mice infected with Rho-DUC or WT were similar (Fig. 6a). The lesions observed in lungs at day 168 post infection were similar for both strains (Fig. 6b). When doxycycline was added to the food of mice infected with Rho-DUC during the acute or the chronic phase of

the infection, CFU counts decreased in lungs and spleens below the limit of detection of one CFU per organ. Furthermore, when silencing of *rho* was initiated after a successful establishment of the infection, the number of lesions observed on the lungs of infected mice was dramatically reduced at day 168 post infection, in agreement with the reduction of CFU counts (Rho-DUC doxycycline day 35, Fig. 6b). For eight of these mice, we removed doxycycline from the food from day 200 onward. When these mice were analysed for relapse 300 days post infection CFU counts were still below the level of detection and the lungs showed no sign of infection (Fig. 6). This is remarkable and demonstrates that Rho is not only critical for *M. tuberculosis* to establish and maintain an infection in mice but also suggest that inhibition of Rho could be sufficient to achieve efficient sterilization.

## Discussion

Inhibiting bacterial transcription with RIF is one of the most efficient strategies to treat infections with *M. tuberculosis*. The transcription termination factor Rho is absent from the genomes of eukaryotes and susceptible to inhibition by BCM. BCM, which is also sold under the trade name bicozamycin, is primarily used in veterinary medicine, but has been shown to be of clinical value for the treatment of diarrhea in humans[43]. The vulnerability of *M. tuberculosis* towards inhibition of transcription and the proven druggability of Rho thus suggest Rho to be an interesting target for TB drug development. However, the importance of Rho for growth and survival varies across the bacterial kingdom, as Rho is generally essential in gram-negative bacteria but often dispensable in gram-positives. Despite being classified as gram-positive, *M. tuberculosis* has been predicted by transposon mutagenesis to require Rho for growth[24,25], but no direct evidence for the essentiality of Rho in *M. tuberculosis* had been shown.

We constructed conditional knockdown mutants to study the consequences of inhibiting Rho-dependent transcription termination and found Rho to be crucial for transcriptional accuracy across the entire genome of *M. tuberculosis*. Depleting Rho for 6 h, while not killing *M. tuberculosis* (Supplementary Fig. 7), was sufficient to significantly change the abundance of >500 transcripts (Supplementary Fig. 3). Transcripts affected by Rho were enriched for genes of two classes (Fig. 3). The first consists of genes related to insertion elements and phages. The silencing of foreign DNA is one of the primary functions of Rho *E. coli* and BCM is less active against an *E. coli* strain that has been cured of all phages[44]. That genes related to insertion elements and phages are among those most susceptible to inhibition of Rho in *M. tuberculosis* suggests its role as a transcriptional immunity factor to be conserved. Furthermore, Rho seems to be more important for silencing gene expression in *M. tuberculosis* than in *E. coli*. In *E. coli*, only 11% of the regions affected by inhibition of Rho affect sense transcripts[14]; for *M. tuberculosis* this number is 28%. The second group of genes that is more frequently affected by inhibition of Rho consists of PE/PPE genes, whose susceptibility to silencing by Rho has not been recognized earlier, as they are unique to mycobacteria. Several PE/PPE genes have been implicated in mycobacterial virulence[38], and the regulated expression of such genes through Rho termination or lack thereof could play a role during infection.

Our studies also clearly show that the importance of Rho in *M. tuberculosis* goes beyond silencing foreign DNA and virulence factors during growth *in vitro*. RSRs are distributed along the entire genome and Rho is required for the accurate transcription of genes belonging to every functional class (Fig. 3). Furthermore, if Rho depletion continues for >6 h the entire genome becomes

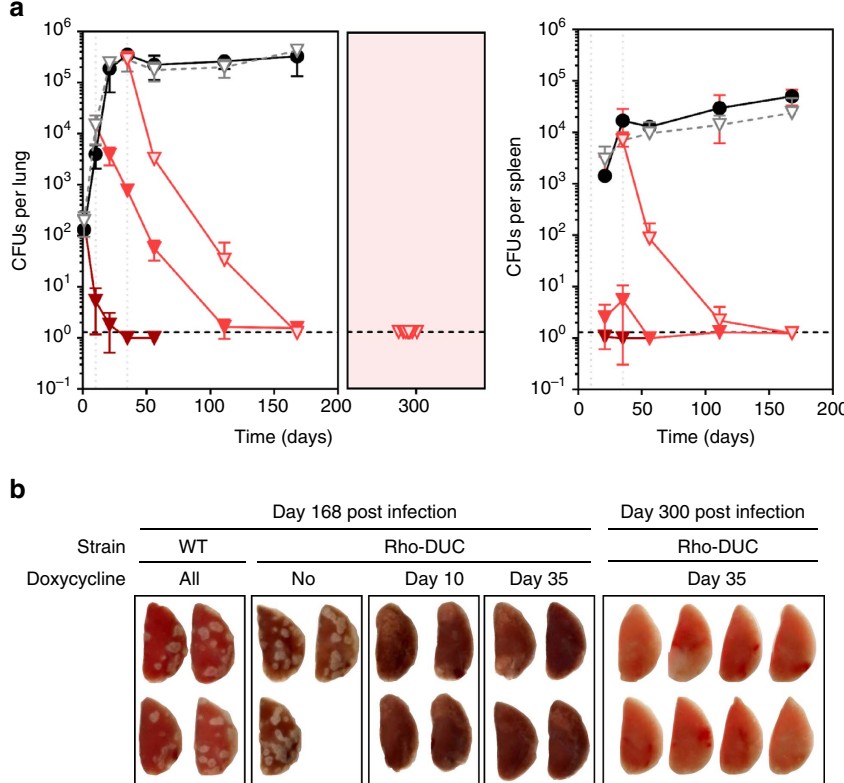

**Figure 6 | Depletion of Rho kills *M. tuberculosis* in mice.** CFU isolated from lungs and spleens of mice infected by WT or Rho-DUC (**a**). The mice infected by the mutant received regular food for the whole duration of the experiment (open grey triangles), or were given food supplemented with doxycycline starting before the infection (dark brown triangles), at day 10 post infection (red triangles) or at day 35 post infection (light pink triangles). The mice infected by the WT Mtb strain received doxycycline-supplemented food for the whole duration of the experiment (black circles). In the pink insert mice that were infected with Rho-DUC received doxycycline-supplemented food from day 35 to day 200, and received doxycycline-free food from day 200 to 300. Data are averages ( ± s.d.) from 3–8 mice as indicated in (**b**), the dashed line represents the limit of detection of 1.3 CFUs per lung or per spleen. (**b**) Lungs collected at day 168 and day 300 post infection from mice infected with the indicated strains.

transcribed, eliciting a dramatic and widespread increase in antisense transcription (Supplementary Fig. 6). Rho is thus crucial to the transcriptional integrity of the entire *M. tuberculosis* genome. It is difficult to envision how the pathogen could survive inactivation of Rho either by physiological adaption or mutations other than those that would prevent inactivation of Rho. In accordance with the transcriptome data, the phenotypic consequences of Rho depletion are immediate and drastic under every condition analysed.

Inactivation and overexpression of Rho prevented *M. tuberculosis* from forming colonies on agar plates (Fig. 1). Silencing was equally cidal during growth and starvation (Fig. 4). This dependency on Rho is remarkable for two reasons, the first being that inactivation of Rho reduced CFU by five orders of magnitude within a week. For comparison, genetic inactivation of many other attractive targets for drug development only decreased CFU by three orders of magnitude or less over a similar time period[34,45–50]. Second, not all genes required for growth are required for non-replicating persistence and drugs, for example INH, which is cidal for growing bacteria, can be virtually inactive against non-replicating bacteria. This antibiotic tolerance induced by a change in growth rate leads to so-called class II persisters, whose slow eradication by the current drugs is thought to contribute to the length of TB chemotherapy[51]. The drastic reduction of viability that is achieved by inactivating Rho suggests that targeting Rho could improve TB chemotherapy by eradicating class II persisters. In accordance with the *in vitro* data, depleting Rho effectively eliminated *M. tuberculosis* during

acute and chronic mouse infection (Fig. 6). Additional experiments will be required to determine the extent to which relapse can be prevented by targeting Rho alone, but it is encouraging that none of the mice analysed showed any sign of relapse 100 days after treatment with doxy had been stopped (Fig. 6).

BCM is not very active against Rho of *M. tuberculosis*[26] and poorly absorbed when given orally. Targeting Rho for TB drug development will thus depend on improving the potency, bioavailability of BCM and possibly specificity or on identifying novel inhibitors. ATPase inhibitors can be efficiently screened and such molecules include already approved drugs[52]. However, the importance of Rho's ATPase activity has not been consistently found to be essential for transcription termination *in vitro*[26,41]. We therefore tested if Rho variants carrying point mutations that interfere with ATP binding could rescue *M. tuberculosis* from Rho-depletion induced growth inhibition. We found these mutated Rho proteins to be expressed but unable to complement for the loss of WT Rho (Fig. 5). Small-molecule inhibitors of Rho's ATPase activity could thus be productive starting points leading to novel drugs. They might, given the drastic impact of Rho inactivation on replicating and non-replicating *M. tuberculosis in vitro* and during infections, replicate the effects of inhibiting RNAP by RIF.

## Methods

**Plasmids and general procedures.** Plasmids are listed in Supplementary Table 1. The plasmids were constructed using Gateway Cloning Technology (Life

technologies) following the manufacturer's instructions. *E. coli* MACH1 (Invitrogen) was used as the cloning host and grown in LB. When necessary, antibiotics were added at the following concentrations: kanamycin 40 µg ml$^{-1}$, zeocin 25 µg ml$^{-1}$, streptomycin 25 µg ml$^{-1}$, and hygromycin 50 µg ml$^{-1}$. Protein lysates were prepared from 30 ml cultures bacterial pellets. The cells were washed in phosphate saline buffer (PBS) containing 0.05% tyloxapol followed by PBS and resuspended in 0.8 ml of PBS containing 1 × protease inhibitor cocktail (Roche). The cells were lysed by bead beating three times 30 s at 4,500 r.p.m. in a mini-beadbeater (Biospec) with 0.1 mm Zirconia/Silica beads. After removing the beads and cell wall by centrifuging, the lysates were filtered through a 0.2 µm SpinX column (Corning) and protein concentration was measured with the Bio-Rad DC Protein Assay Kit. Proteins were separated on a SDS–polyacrylamide gel electrophoresis (SDS–PAGE) gel and transferred onto a nitrocellulose membrane before being probed by a dilution in Odyssey buffer (LI-COR Biosciences) of rabbit serum raised against purified *M. tuberculosis* Rho (obtained from Thermo Fisher), against Dlat (ref. 53) or an anti-FLAG antibody (Sigma) followed by detection with a fluorescent secondary antibody (LI-COR Biosciences).

**Mutant construction and characterization.** Strains used and generated in this study are listed in Supplementary Table 2. The *M. tuberculosis* H37Rv (gift from Robert North, Trudeau Institute) strain harbouring a plasmid encoding the enzymes required for recombineering (pNI-Rec-ET (ref. 54)) was transformed with pGMCS-Prho to create the merodiploid strain expressing a second copy of *rho*. Next, the native copy of *rho* was inactivated by homologous recombination using a hygromycin resistance cassette as the antibiotic marker. The genotype of the mutant at the *rho* locus was verified by specific PCRs, southern blot and sequencing of the regions flanking the integration site (Supplementary Fig. 1). After curing of pNI-Rec-ET, the att-L5 complementation construct was swapped out by transformations with plasmids carrying alternate antibiotic resistance markers. In the Rho-DUC strain, the DAS + 4-tagged copy of *rho* and the transcriptional repressor revTetR are expressed from the *att-L5* site and sspB together with its expression regulation system are expressed from the constructed integrated at the attachment site of the *Tweety* phage. When this strain is additionally complemented, the plasmid is integrated into attachment site of the phage *Giles*.

Colonies of *M. tuberculosis* were grown at 37 °C on Middlebrook 7H11 agar supplemented with 0.5% glycerol and 10% oleic acid-albumin-dextrose-catalase (OADC Becton Dickinson). When indicated, atc was added on a paper disc (100 ng). Replicating cultures of *M. tuberculosis* Rho-DUC strains and derivatives were incubated at 37 °C in Middlebrook 7H9 medium with 0.5% glycerol, 0.02% tyloxapol (Sigma), 0.5% BSA (Roche), 0.2% dextrose, 0.085% NaCl and the appropriate antibiotics at the following concentrations: kanamycin at 20 µg ml$^{-1}$, streptomycin at 20 µg ml$^{-1}$, zeocin at 25 µg ml$^{-1}$ and hygromycin at 50 µg ml$^{-1}$. When indicated atc was added at the concentration of 1 µg ml$^{-1}$. The procedure for cultivating the Tet-ON strains was identical except the precultures incubated in the presence of 500 ng ml$^{-1}$ were washed twice in PBS before being used to start the cultures in the presence or absence of atc at 1 µg ml$^{-1}$. The non-replicating cultures of *M. tuberculosis* were obtained by washing 7H9 grown bacteria twice in PBS containing 0.05% tyloxapol and resuspending them in PBS at OD600 nm of 1. When applicable the antibiotics were added at the concentrations indicated above for replicating conditions. After 10 days, atc was added to half of the cultures at the concentration of 1.5 µg ml$^{-1}$.

The control strain used with the Rho-DUC strain in both replicating and non-replicating experiments and in the mice infection was the WT strain harbouring the plasmids pGMCZq1-T38S38-0X and pGMCtKq27-TSC10M1-sspB respectively integrated into the *att-L5* and the *tweety* sites.

**Mouse infections.** Animal studies were carried out in accordance with the guide for the fare and use of Laboratory Animals of the National Institutes of Health, with approval from the Institutional Animal Care and Use Committee of Weill Cornell Medical College. All animals were maintained under specific pathogen-free conditions and fed water and chow ad libitum, and all efforts were made to minimize suffering or discomfort. Female 8 week-old female C57BL/6 (Jackson Laboratory) were infected using an inhalation exposure system (Glas-Col) with early-log-phase *M. tuberculosis* to deliver ∼100 bacteria per mouse. Doxycycline containing food (2,000 p.p.m., Research Diets) was given to mice starting at the indicated time-points. Four mice were killed at each time point for each condition, except for the day 294 when eight mice were killed and the day 168 when only three mice infected with Rho-DUC strain were killed. Bacterial load in the lungs and in the spleen were determined by plating serial dilution of the organs homogenates. Upper left lung lobes were fixed in 10% buffered formalin, embedded in paraffin and stained with hematoxylin and eosin.

**RNA extraction and cDNA library preparation.** Cultures of *M. tuberculosis* were mixed with an equal volume of GTC buffer containing guanidinium thiocyanate (5 M), sodium N-lauroylsarcosine (0.5%), Trisodium citrate dihydrate (25 mM) and 2-mercaptoethanol (0.1 M) and pelleted by centrifugation. Pellets were resuspended in Trizol and frozen at − 80 °C. After thawing, zirconium beads were added and the samples were bead-beaten for three 60 s pulses alternated with 60 s incubations on ice. Supernatants were mixed with chloroform and centrifuged at

4 °C for 10 min. Total RNAs were purified from the upper phase using the Direct-zol RNA miniprep kit (Zymo) according to manufacturer's instructions. RNA quantity and integrity were determined from agarose gel, Qubit assay (Thermo-Fisher Scientific) and electropherograms (Agilent Bioanalyzer). Illumina complementary DNA(cDNA) libraries were generated using the RNAtag-Seq protocol as described[55]. Briefly, 1 µg of total RNA was fragmented, depleted of genomic DNA, dephosphorylated, then ligated to DNA adaptors carrying 5′-AN8-3′ barcodes with a 5′ phosphate and a 3′ blocking group. Barcoded RNAs were pooled and depleted of ribosomal RNA (rRNA) using the RiboZero rRNA depletion kit (Epicentre). These pools of barcoded RNAs were converted to Illumina cDNA libraries in three main steps: (i) reverse transcription of the RNA using a primer designed to the constant region of the barcoded adaptor; (ii) degradation of the RNA and ligation of a second adaptor to the single-stranded cDNA; (iii) PCR amplification using primers that target the constant regions of the 3′ and 5′ ligated adaptors and contain the full sequence of the Illumina sequencing adaptors. cDNA libraries were sequenced on Illumina HiSeq 2,000 to generate 25 base paired-end reads. The second read generated from RNAtag-Seq libraries is derived from the barcoded adaptor ligated directly to the 3′ nucleotide of RNAs, enabling high resolution mapping of 3′ ends. For each condition, 3–4 replicates were processed in two rounds of library preparation and sequencing except for the Rho-DUC incubated with atc for 24 h for which duplicates samples were processed in the same library. Metrics of the RNAseq samples are provided in Supplementary Data 8.

**RNA-Seq data analysis.** For the analyses of RNAtag-Seq data, reads from each sample in the pool were identified based on their associated barcode using custom scripts, and up to one mismatch in the barcode was allowed with the caveat that it did not enable assignment to more than one barcode. Barcode sequences were removed from reads, and the reads from each sample were aligned to genes in their cognate strain using BWA (ref. 56). Gene annotations were obtained from Tuberculist (accession AL123456.3) and reads per gene were calculated using custom scripts. Differential expression was measured using DESeq2 package of Bioconductor[57]. RNA-Seq data were visualized on the Artemis browser[58]. Matlab was used to define RSRs as illustrated in Supplementary Figs 4 and 5.

**Classification of RSRs.** We classified RSR according to schema shown in Supplementary Fig. 10. The intragenic RSRs (class A in Supplementary Fig. 10) that covered >50% of a gene were scored as silencers, whereas intragenic RSRs that covered <50% of a gene but included the stop codon were categorized as terminators. To classify intergenic RSRs, we first grouped them into three classes (B, C and D) illustrated in Supplementary Fig. 6. For RSRs of classes B and C, we computed the distances between the first nucleotide of the RSR and either the last nucleotide of the upstream gene (Distance to upstream Gene, DupG) or the first nucleotide of the downstream gene (Distance to downstream Gene, DdwG) (Supplementary Fig. 10). The median values for DupG and DdwG were 600 and 425 nucleotides, respectively. Further, we categorized as terminators all class B RSRs with a DupG of <600 and as silencers all class C RSRs that extended of more than half of the downstream gene, and had a DdwG of <425 nucleotides. The remaining intergenic RSRs were classified in class D with no relation to sense genes. All classes RSRs were found evenly distributed on both strands along the chromosome (Supplementary Fig. 11). Examples of these classes are supplied in Supplementary Fig. 10.

**Identification of Rho-independent termination.** For each nucleotide position P of the transcriptome of the strains WT 0 h, WT 6 h and DUC 0 h, using Matlab, we computed the slope (S) of a theoretical line $y = ax$ between the position P and P + 5 (S1), P and P + 15 (S2), and P and P + 50 (S3). A position P was defined as the first nucleotide of an intrinsic terminator if S3 ≤ 0, S1 < S2 < S3 and S1 < − 0.01. To select for strong terminators we only chose the top 25% hits with a highest expression ratio (P − 100, P)/(P + 15, P + 100) for final analyses. The lists obtained from each sample WT 0 h, WT 6 h, and DUC 0 h have been cross-compared to keep the hits found <10 bp apart in the three conditions.

***In silico* Rho-independent terminator prediction.** *In silico* predictions of Rho-independent terminators were generated using (1) RNAMotif v3.0.4 (ref. 59) with a motif descriptor (Supplementary Data 9) provided by David J. Ecker (Ibis Therapeutics, Carlsbad CA, USA), (2) TransTermHP v2.05 (ref. 60), (3) FindTerm, a program created by Gilgi Friedlander based on a heuristic algorithm developed by Ruth Hershberg[61], and (4) RNIE (ref. 37). Predicted Rho-independent terminators were also obtained from Mitra *et al.*[36]

**Data availability.** Sequencing data that support the findings of this study have been deposited in the Sequence Read Archive (SRA) under BioProject ID: PRJNA354066. The authors declare that all other data supporting the findings of this study are available within the article and its Supplementary Information Files, or from the corresponding author on request.

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

## Acknowledgements

This work was supported by grants from the NIH (U19AI111143, Tri-Institutional TB Research Unit, part of the NIAID TBRU Network) and the Bill & Melinda Gates Foundation (OPP1024065, part of the TB Drug Accelerator). We thank Christopher Sassetti and Kenan Murphy (University of Massachusetts) for pNIT-Rec-ET, Carl Nathan (Weill Cornell Medicine) for Dlat-specific antiserum and PrcB-specific antiserum, Shuang Song (Weill Cornell Medicine) for assistance with purifying Rho and Sinéad Chapman (Broad Institute of MIT and Harvard) for her help in submitting sequencing data to NCBI.

## Author contributions

L.B. and J.L. performed experiments. L.B., J.V. and J.L. performed analyses. D.S. and L.B. designed the experiments. L.B., J.V. and D.S. wrote the paper, which was edited by J.L.

## Additional information

**Competing interests:** The authors declare no competing financial interests.

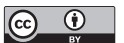

