## [Peer Review File · Nature Communications]

Reviewers' comments:

Reviewer #1 (Remarks to the Author):

In this Ms, Botella et al. describe extensive, seminal work on the role and importance of transcription termination factor Rho in *Mycobacterium tuberculosis* (Mtb hereafter). Using conditional knockdown mutants, they provide the first line of direct evidence showing that Rho is essential for Mtb viability and infectivity. Using RNAseq, they delineate, for the first time, the pool of Rho targets in Mtb and notably show that Rho controls spurious transcription genome-wide (as observed in other, phylogenetically distinct bacteria) as well as transcription of mycobacteria-specific genes (some of which have been implicated in mycobacterial virulence). They also provide conclusive evidence that the ATPase activity of Mtb's Rho is important for its function in vivo, supporting a commonality of mechanism across species. Finally, the authors provide exciting preliminary data suggesting that rho inhibition could be a very effective strategy against TB. The Ms is generally well written and easy to comprehend with the exception of the sections devoted to RNAseq which would benefit from simplification. Overall, I find the work important and novel, warranting publication in Nature Com. once the points below have been addressed.

1) My main reservation is related to the fact that Rho is overexpressed in the RhoDUC strain as compared to WT (in the reference condition, i.e. absence of atc) while Rho appears to be present in comparable amounts in both strains after 6h exposure with atc (fig. 1C). Therefore, the state of "Rho depletion" evoked by the authors is by comparison with an artificial situation (overexpression in absence of atc) and this should be made clearer throughout the Ms. While Rho overexpression in RhoDUC does not appear to affect the transcriptome very much when compared to that of WT(t0), it is surprising (and a bit disturbing) to see such dramatic changes in RhoDUC after 6h exposure with atc (>500 transcripts affected) while the WT transcriptome is hardly changed. How do the authors reconcile these observations with the fact that Rho levels are comparable in RhoDUC and WT under this condition (fig. 1C)?

2) I find the classification of RSRs (and explanations given) a bit confusing (except for class D). Most RSRs are likely to stem from the inactivation of a Rho-dependent termination event occurring at a site (Rho-dependent terminator) which, however, may be far from the actual start of the RSR (due, for instance, to exonucleolytic trimming of the transcripts). Hence, the actual Rho-dependent terminators and mRNA terminators evoked in the text are distinct notions which might be easily confused by the average reader. I also find subdividing classes A, B, and C on arbitrary thresholds unnecessary and confusing and suggest to simplify this section of the Ms significantly.

3) I am not sure of the protocol used to detect intrinsic (Rho-independent) terminators; precise detection often relies on R2 reads from paired-end Illumina sequencing. Does the protocol used here allow the same kind of precision (more details in methods would be helpful)? If so, is there any consensus feature (e.g. GC-rich hairpin, U-tract) for the Mtb intrinsic terminators (or classes thereof) identified here? Providing sequence logos might be helpful. Do the candidate terminators include those tested experimentally by others (e.g. Czyz et al., *Mbio* 2014 Apr 8;5(2):e00931)? This would help validate the author's approach and compensate for the lack of independent (alternative) testing of the candidate terminators.

4) Lane 63: factor-dependent terminators are not exclusive to prokaryotes. There are many examples in eukaryotes, one being termination of RNA pol II transcription mediated by Sen1, yeast's functional homolog of Rho.

5) Lanes 65-67: In the most recent work (ref 33), ~1300 Rho-dependent loci rather than ~200 loci (ref 13) have been detected in *E. coli*.

6) Lane 152: I guess the authors mean "Rho-independent" rather than "Rho-dependent" here.

7) Lanes 218-219: might be worth precisizing that the mutated residues are located in the Walker B and ATP finger motifs critical for ATP hydrolysis.

8) Lane 266: ref 33 missing.

9) Lane 588: typo or word missing.

10) Figure 1C: a quantitative representation of the relative amounts of Rho would be helpful.

11) Figure 6: open and light pink symbols are poorly distinguished.

- 12) Figure S1: unclear if the whole region depicted in (a) is that which has been integrated in attL5. Panel (d) is mislabeled.
- 13) Figure S2: it would be helpful to have the histograms for Rho-DUC 24h also shown here.
- 14) Figure S6: I find panel (c) rather cryptic. The authors should explore alternative representations.

Reviewer #2 (Remarks to the Author):

In this work Botella and colleagues analyze the impact of the depletion of Rho on *Mycobacterium tuberculosis* viability in vitro and in vivo, and on the global transcription profile. The study reveals that Rho acts as a terminator and as a silencer, while its inactivation causes increased expression of antisense transcripts and loss of viability. By means of conditional expression systems, the Authors elegantly demonstrate the potential for Rho as a drug target, whose inactivation rapidly leads to bacterial clearance from the infected animals.

The manuscript is well written, experiments seem to have been conducted carefully, data are well presented. Overall, the work complements existing studies and improves our understanding of the *M. tuberculosis* transcriptome.

I do not have any major criticism except for the following points:

1. As I said above, the article is well written but I found it a bit «sterile», especially where the RNAseq data are presented. Plenty of information is included in the supplementary tables, however such a piece of work should mention more examples in the Results section. Please see my comments below as well.
2. The Authors frequently mention the «transcriptional accuracy» which is compromised upon depletion of Rho. They should clarify what «accuracy» means. I interpreted it as the ability, or lack thereof, of RNA polymerase to start and stop transcription at the appropriate genomic position. Is this correct?
3. Results section: «...a failure of transcription termination would result in an increase in transcript abundance». In my opinion, failure in transcription termination should also correspond to presence of longer transcripts. This should be pointed out and examples of longer transcripts provided. For instance, the Authors could refer to the existing predicted operons and check whether these gave rise to longer transcripts when rho was silenced.
4. Are there any transcripts initiated in response to Rho depletion? Have these been considered as part of the RSRs (class D)? Again, I would appreciate some examples for each class of RSR.
5. Supplementary Table 8. List of RIT. It would be useful to add the names of the closest genes.
6. The Authors should elaborate more on the 113 RSRs located in regions that do not contain ORFs. What are these RSRs? Small RNAs? Insertion sequences? New features not yet annotated?
7. I found the silencing of PE/PPE proteins intriguing. Could these be considered as foreign DNA? Horizontal gene transfer in *M. tuberculosis* has not been proven yet, however I find it fascinating that this class of proteins could have arisen from HGT followed by gene duplication events. Could the Authors comment on this point?
8. There is a conflict between the names given to the Excel sheets and the list of Supplementary Tables in the PDF file that contains the Supplementary Material. Please fix this point.

Reviewer #3 (Remarks to the Author):

The authors investigate the importance of the transcription termination protein, Rho, in survival and global transcription of *Mycobacterium tuberculosis*. They conclude that Rho is essential, with Rho depletion leading to loss of viability both in culture and during a mouse infection. Loss of Rho is associated with large changes in global transcription, due to failure to terminate canonical and non-canonical RNAs.

The work is extremely thorough and the data are of a universally high quality. Moreover, the conclusions are all fully justified. However, the manuscript doesn't do much to advance the field. Rho was already established as an essential gene in *M. tuberculosis*, and the effect of Rho depletion on global transcription is essentially the same as it is in *E. coli* and *B. subtilis*. In other words, the manuscript confirms what we already knew or strongly suspected.

Reviewer #1 (Remarks to the Author):

In this Ms, Botella et al. describe extensive, seminal work on the role and importance of transcription termination factor Rho in *Mycobacterium tuberculosis* (Mtb hereafter). Using conditional knockdown mutants, they provide the first line of direct evidence showing that Rho is essential for Mtb viability and infectivity. Using RNAseq, they delineate, for the first time, the pool of Rho targets in Mtb and notably show that Rho controls spurious transcription genome-wide (as observed in other, phylogenetically distinct bacteria) as well as transcription of mycobacteria-specific genes (some of which have been implicated in mycobacterial virulence). They also provide conclusive evidence that the ATPase activity of Mtb's Rho is important for its function in vivo, supporting a commonality of mechanism across species. Finally, the authors provide exciting preliminary data suggesting that rho inhibition could be a very effective strategy against TB. The Ms is generally well written and easy to comprehend with the exception of the sections devoted to RNAseq which would benefit from simplification. Overall, I find the work important and novel, warranting publication in Nature Com. once the points below have been addressed.

We very much thank the reviewer for the appreciation and enthusiasm.

1) My main reservation is related to the fact that Rho is overexpressed in the RhoDUC strain as compared to WT (in the reference condition, i.e. absence of atc) while Rho appears to be present in comparable amounts in both strains after 6h exposure with atc (fig. 1C). Therefore, the state of "Rho depletion" evoked by the authors is by comparison with an artificial situation (overexpression in absence of atc) and this should be made clearer throughout the Ms. While Rho overexpression in RhoDUC does not appear to affect the transcriptome very much when compared to that of WT(t0), it is surprising (and a bit disturbing) to see such dramatic changes in RhoDUC after 6h exposure with atc (>500 transcripts affected) while the WT transcriptome is hardly changed. How do the authors reconcile these observations with the fact that Rho levels are comparable in RhoDUC and WT under this condition (fig. 1C)?

We thank the reviewer for giving us an opportunity to clarify this point. In the revised manuscript we added new data (Supplementary Fig. 2), which indicate that Rho-DAS has a lower activity *in vivo* than WT Rho. This likely explains why Rho-DAS is required at higher expression levels than WT Rho to achieve normal growth and WT-like Rho-dependent transcription termination.

2) I find the classification of RSRs (and explanations given) a bit confusing (except for class D). Most RSRs are likely to stem from the inactivation of a Rho-dependent termination event occurring at a site (Rho-dependent terminator) which, however, may be far from the actual start of the RSR (due, for instance, to exonucleolytic trimming of the transcripts). Hence, the actual Rho-dependent terminators and mRNA terminators evoked in the text are distinct notions which might be easily confused by the average reader. I also find subdividing classes A, B, and C on arbitrary thresholds unnecessary and confusing and suggest to simplify this section of the Ms significantly.

We agree with reviewer 1 and appreciate his/her comments. To improve the manuscript accordingly, we state more clearly in the revised manuscript that RSRs don't identify Rho-dependent terminators directly (lines 117 - 120). We also avoid (so we believe) any inappropriate use of the term "Rho-dependent terminator" throughout the manuscript and have improved the terminology when we discuss the function of RSR as silencers.

Furthermore, we removed references to the different classes of RSRs from the main text and simplified the respective section of the main text considerably (the respective paragraph starts with line 148). We kept the figure that illustrates these four RSR classes in the manuscript because it helps to document how we defined the RSRs that we discuss as silencers.

3) I am not sure of the protocol used to detect intrinsic (Rho-independent) terminators; precise detection often relies on R2 reads from paired-end Illumina sequencing. Does the protocol used here allow the same kind of precision (more details in methods would be helpful)? If so, is there any consensus feature (e.g. GC-rich hairpin, U-tract) for the Mtub intrinsic terminators (or classes thereof) identified here? Providing sequence logos might be helpful. Do the candidate terminators include those tested experimentally by others (e.g. Czyz et al., *Mbio* 2014 Apr 8;5(2):e00931)? This would help validate the author's approach and compensate for the lack of independent (alternative) testing of the candidate terminators.

The sequence analyses that we performed did not reveal conserved consensus features. The four intrinsic terminators that Czyz et al. confirmed to be functional *in vitro* included a sequence located directly downstream of *tuf* (rv0685). *Tuf* was also identified by the criteria we used to identify regions affected by Rho-independent termination.

We agree with the reviewer that we cannot claim to have identified new Rho-independent terminators (RITs) and that the regions we identified likely include mechanistically diverse elements. We there modified our discussion of these elements (lines 143-145, line 385 and title of Supplementary table 4)

However, it seems worth pointing out that we identifying new RITs was not a primary goal of this work. Instead we performed this analysis to assure ourselves that addition of *atc* to the Rho-DUC mutant did not cause general and unspecific effects on transcription termination. We believe that the analysis we performed was sufficient to achieve this goal.

4) Lane 63: factor-dependent terminators are not exclusive to prokaryotes. There are many examples in eukaryotes, one being termination of RNA pol II transcription mediated by Sen1, yeast's functional homolog of Rho.

Thank you for preventing us of this omission. The text has been edited accordingly.

5) Lanes 65-67: In the most recent work (ref 33), ~1300 Rho-dependent loci rather than ~200 loci (ref 13) have been detected in *E. coli*.

This has been corrected.

6) Lane 152: I guess the authors mean "Rho-independent" rather than "Rho-dependent" here.

We agree that this sentence was confusing and removed it from the revised manuscript.

7) Lanes 218-219: might be worth precisizing that the mutated residues are located in the Walker B and ATP finger motifs critical for ATP hydrolysis.

We have added this information in lines 193 – 196 of the revised manuscript.

In addition, while this manuscript was under review, we also evaluated the ability of the Rho E386A to complement Rho deletion in *M. tuberculosis*. *In vitro*, the Rho E386A variant has no ATPase or helicase activities and fails to terminate transcription. Importantly, this has been shown for both Rho from *E. coli* and Rho from *M. tuberculosis* (Balasubramanian and Stitt, 2010, *J Mol Biol* 404; D'Heygere et al., 2015, *Nucleic Acid Res*, 43). In the new Supplementary Figure 9, we

demonstrate that Rho E386A cannot complement the growth defect of Rho-DUC mutant grown in a medium supplemented with atc.

8) Lane 266: ref 33 missing.

Thank you for noticing this. The reference has been inserted.

9) Lane 588: typo or word missing.

This mistake has been corrected.

10) Figure 1C: a quantitative representation of the relative amounts of Rho would be helpful.

For the reasons described in our answer in 1, we have opted for not doing so.

11) Figure 6: open and light pink symbols are poorly distinguished.

The symbols color has been changed.

12) Figure S1: unclear if the whole region depicted in (a) is that which has been integrated in attL5. Panel (d) is mislabeled.

For clarity, the labels on the figures have been modified. Panel (d) has been re-labeled.

13) Figure S2: it would be helpful to have the histograms for Rho-DUC 24h also shown here.

The analysis shown in this figure required 3-4 replicate experiments per time point (processed in two independent rounds of libraries preparation and sequencing). The number of CFUs of Rho-DUC strain dramatically declines of 2.5 logs after 24 hours of Rho depletion (Supplementary Figure 7) and for this reason we did not include this time point in our later RNAseq experiments. We have thus for the 24h time point only 2 replicates which were processed in the same library and sequenced in the same run and we excluded the condition from this statistical analysis. We clarified the number of replicates performed per time point in the revised methods section (lines 358 to 361).

14) Figure S6: I find panel (c) rather cryptic. The authors should explore alternative representations.

We agree that the panel c of Supplementary Fig. 6 was too complex. In the revised Figure (Supplementary Fig. 6), we simplified it as follows: panels a and b report the concomitant increase of the number of reads attributed to antisense genomic features and a greater coverage of the chromosome at the nucleotide level as a consequence of a failure of transcription termination. Additionally, the panel c shows that *rho* silencing over a longer period led to a greater number of nucleotides with higher normalized reads.

Reviewer #2 (Remarks to the Author):

In this work Botella and colleagues analyze the impact of the depletion of Rho on *Mycobacterium tuberculosis* viability *in vitro* and *in vivo*, and on the global transcription profile. The study reveals that Rho acts as a terminator and as a silencer, while its inactivation causes increased expression of antisense transcripts and loss of viability. By means of conditional expression systems, the Authors

elegantly demonstrate the potential for Rho as a drug target, whose inactivation rapidly leads to bacterial clearance from the infected animals.

The manuscript is well written, experiments seem to have been conducted carefully, data are well presented. Overall, the work complements existing studies and improves our understanding of the *M. tuberculosis* transcriptome.

Thank you for your appreciation and enthusiasm.

I do not have any major criticism except for the following points:
1. As I said above, the article is well written but I found it a bit «sterile», especially where the RNAseq data are presented. Plenty of information is included in the supplementary tables, however such a piece of work should mention more examples in the Results section. Please see my comments below as well.

2. The Authors frequently mention the «transcriptional accuracy» which is compromised upon depletion of Rho. They should clarify what «accuracy» means. I interpreted it as the ability, or lack thereof, of RNA polymerase to start and stop transcription at the appropriate genomic position. Is this correct?

This is correct. We clarified this point in the revised version of the paper: “Prolonged depletion of Rho thus caused transcriptional accuracy to collapse and RNA synthesis to continue, when it would have otherwise been halted, thus leading to the accumulation of pervasive antisense transcripts.” (lines 168 - 170)

3. Results section: «...a failure of transcription termination would result in an increase in transcript abundance». In my opinion, failure in transcription termination should also correspond to presence of longer transcripts. This should be pointed out and examples of longer transcripts provided. For instance, the Authors could refer to the existing predicted operons and check whether these gave rise to longer transcripts when rho was silenced.

We agree and have modified the main text as follows: “We anticipated that a failure of transcription termination would increase transcript length (Supplementary Fig. 4a). We therefore focused our analyses on the 6h time point when 96% of the changes affecting sense transcripts stemmed from areas of the genome that were underrepresented in the transcriptome before Rho was depleted (Supplementary Fig. 3a).” (lines 114 - 117)

It is important to note that RNAseq cannot measure transcript length directly, but our analysis was designed to enrich for changes of the transcriptome that were due to changes in transcript elongation. The RSRs reported in Fig. 2 likely are examples of longer transcripts as reads are detected over a longer genomic region when Rho is depleted.

4. Are there any transcripts initiated in response to Rho depletion? Have these been considered as part of the RSRs (class D)? Again, I would appreciate some examples for each class of RSR.

Among the 113 RSRs located in regions that do not contain ORFs on the same DNAS strand, some transcripts may indeed have been initiated in response to Rho depletion. However, our RNAseq data do not allow to directly distinguish between initiation of new transcripts and other changes that can affect transcript abundance. Clearly, the widespread and complex changes that occur in the transcriptome after prolonged depletion of Rho are likely caused by multiple mechanisms, which include the activation of promoters and initiation of new transcripts.

Revised Supplementary Fig. 10 includes examples for all classes of RSRs.

5. Supplementary Table 8. List of RIT. It would be useful to add the names of the closest genes.

This has been added.

6. The Authors should elaborate more on the 113 RSRs located in regions that do not contain ORFs. What are these RSRs? Small RNAs? Insertion sequences? New features not yet annotated?

In the revised manuscript we emphasize more clearly that these RSRs are located in regions that do not contain ORFs on the same strand, but frequently do contain ORFs on the other strand. Thus, these RSRs are a reason for the drastic increase in antisense RNA that occurs after inactivation of Rho.

We also carefully analyzed the regions containing these 113 RSRs by manual inspections. As far as we can tell, these do not encode for small RNAs nor do they contain insertion sequences. As suggested by reviewer 2, they may be new features of unknown function.

7. I found the silencing of PE/PPE proteins intriguing. Could these be considered as foreign DNA? Horizontal gene transfer in *M. tuberculosis* has not been proven yet, however I find it fascinating that this class of proteins could have arisen from HGT followed by gene duplication events. Could the Authors comment on this point?

We agree. This is a fascinating hypothesis and thank the reviewer for pointing it out to us. However, we don't think that there is yet enough evidence to discuss this hypothesis in the manuscript.

8. There is a conflict between the names given to the Excel sheets and the list of Supplementary Tables in the PDF file that contains the Supplementary Material. Please fix this point.

Thank you for pointing this out. The errors have been corrected.

Reviewer #3 (Remarks to the Author):

The authors investigate the importance of the transcription termination protein, Rho, in survival and global transcription of *Mycobacterium tuberculosis*. They conclude that Rho is essential, with Rho depletion leading to loss of viability both in culture and during a mouse infection. Loss of Rho is associated with large changes in global transcription, due to failure to terminate canonical and non-canonical RNAs.

The work is extremely thorough and the data are of a universally high quality. Moreover, the conclusions are all fully justified.

We very much thank reviewer 3 for this assessment.

However, the manuscript doesn't do much to advance the field. Rho was already established as an essential gene in *M. tuberculosis*, and the effect of Rho depletion on global transcription is essentially the same as it is in *E. coli* and *B. subtilis*. In other words, the manuscript confirms what we already knew or strongly suspected.

We respectfully disagree.

It is correct that Rho has been predicted to be essential for optimal growth, but this is only a prediction and this only applied to growth on solid medium. Before this study, it remained unknown if (1) the Tn-seq prediction was indeed correct; (2) if essentiality extended to conditions other than growth on a solid medium; (3) if inactivation of Rho would slow growth, prevent growth, or cause death (all of which would be consistent with the Tn-seq prediction); (4) if *M. tuberculosis* requires Rho to persist in a nonreplicating state (as many genes that are required for growth are dispensable for survival in a nonreplicating state).

Furthermore, because Rho-dependent terminators lack a readily identifiable consensus sequence it was impossible to predict which genes or how many genes would be affected by the inactivation of Rho. In this context, it seems worth noting that the two species that reviewer 3 mentioned (*E. coli* and *B. subtilis*) utilize Rho quite differently as is evident by essentiality of *rho* in one species and not the other.

We demonstrated that inactivation of Rho decreases viability of *M. tuberculosis* during growth *in vitro*, during nonreplicating persistence *in vitro*, and during acute and chronic infection. This decrease in viability is rapid and as drastic (or more drastic) as has been for any gene analyzed in *M. tuberculosis* so far. Cidal activity is likely caused by dramatic and genome-wide changes in transcription and thus unlikely to be susceptible to trivial resistance mechanisms. We furthermore demonstrated that targeting the ATPase activity — which is often druggable — is sufficient to inactivate Rho in *M. tuberculosis*. Collectively, these results define Rho as a highly attractive target for drug development and are very relevant as the belief that all genes predicted to be essential for optimal growth are equally suited for drug development is one of the reasons target-based drug discovery has often failed to deliver new antibiotics.

REVIEWERS' COMMENTS:

Reviewer #1 (Remarks to the Author):

In their revised manuscript, Botella et al. have adequately addressed the (minor) points raised in my initial review of their work

Perhaps in line 195, they should make clearer that mutations in *E. coli* will bear different amino acid numbers. In line 201, they should also make clear which are the two point mutants discussed (since there are now 3 mutants described in the work).

Reviewer #2 (Remarks to the Author):

This is the revised version of a manuscript that I reviewed before.

I carefully read the response of the Authors to my comments and criticisms and found their answers satisfactory.

I do not have any other comment.

Reviewer #1 (Remarks to the Author):

In their revised manuscript, Botella et al. have adequately addressed the (minor) points raised in my initial review of their work.

Thank you again for the constructive criticism of our manuscript.

Perhaps in line 195, they should make clearer that mutations in E. coli will bear different amino acid numbers.

We agree. This could be made clearer and the respective sentence was modified as suggested.

In line 201, they should also make clear which are the two point mutants discussed (since there are now 3 mutants described in the work).

Thank you for pointing this out. The sentence was modified to reflect that protein levels were analyzed for all three mutants.

Reviewer #2 (Remarks to the Author):

This is the revised version of a manuscript that I reviewed before. I carefully read the response of the Authors to my comments and criticisms and found their answers satisfactory. I do not have any other comment.

Thank you again for the careful review of our manuscript.